# Hydrophobic AEROSIL^®^R972 Fumed Silica Nanoparticles Incorporated Monolithic Nano-Columns for Small Molecule and Protein Separation by Nano-Liquid Chromatography

**DOI:** 10.3390/molecules27072306

**Published:** 2022-04-01

**Authors:** Cemil Aydoğan, İbrahim Y. Erdoğan, Ziad El-Rassi

**Affiliations:** 1Food Analysis and Research Laboratory, Bingöl University, Bingöl 12000, Turkey; 2Department of Chemistry, Bingöl University, Bingöl 12000, Turkey; ierdogan@bingol.edu.tr; 3Department of Food Engineering, Bingöl University, Bingöl 12000, Turkey; 4Faculty of Health Sciences, Bingöl University, Bingöl 12000, Turkey; 5Department of Chemistry, Oklahoma State University, Stillwater, OK 74078, USA; elrassi@okstate.edu

**Keywords:** fumed silica, proteomic, monolith, nano-LC, reversed-phase

## Abstract

A new feature of hydrophobic fumed silica nanoparticles (HFSNPs) when they apply to the preparation of monolithic nano-columns using narrow monolithic fused silica capillary columns (e.g., 50-µm inner diameter) was presented. The monolithic nano-columns were synthesized by an in-situ polymerization using butyl methacrylate (BMA) and ethylene dimethacrylate (EDMA) at various concentrations of AEROSIL^®^R972, called HFSNPs. Dimethyl formamide (DMF) and water were used as the porogenic solvents. These columns (referred to as HFSNP monoliths) were successfully characterized by using scanning electron microscopy (SEM) and reversed-phase nano-LC using alkylbenzenes and polyaromatic hydrocarbons as solute probes. The reproducibility values based on run-to-run, column-to-column and batch-to-batch were found as 2.3%, 2.48% and 2.99% (*n* = 3), respectively. The optimized column also indicated promising hydrophobic interactions under reversed-phase conditions, while the feasibility of the column allowed high efficiency and high throughput nano-LC separations. The potential of the final HFSNP monolith in relation to intact protein separation was successfully demonstrated using six intact proteins, including ribonuclease A, cytochrome C, carbonic anhydrase isozyme II, lysozyme, myoglobin, and α-chymotrypsinogen A in nano-LC. The results showed that HFSNP-based monolithic nanocolumns are promising materials and are powerful tools for sensitive separations.

## 1. Introduction

The use of columns with narrower i.d. has been an on-going focus in analytical separation science for several decades. Down-sizing of separation columns have also attracted increasing attention due to their performance for high resolution analyses and point-of-care applications [1,2,3,4]. In this sense, the nano-LC technique is an indispensable analytical tool in bioanalytical separation science and technology, especially in the field of omics research [5]. Various column types, including open-tubular [6], monolithic or packed columns [7], have been used in nano-LC, allowing the highest level of sensitivity [8]. The internal diameters of these columns are between 5–100 µm, which is suitable for Nano-LC [9]. The use of such columns results in an increase in sensitivity, allowing a higher number of molecules to be detected. Regarding multiomics applications, the nanocolumns have found a wide application area in sensitive bioanalytical nano-LC methods [7]. Among others, monolithic columns have been used for the separation of small molecules [10] to very large biomolecules [11] as well as microorganisms [12]. The development of new monolithic column types (e.g., nanoparticle incorporation) may be a key tool in a wide range of cutting-edge research areas. Pillar arrays or packed columns are still the most popular choices. Monolithic columns could provide enormous versatility and can be used for a number of challenging applications.

Nanoparticles (NPs) as a class of substances of particles with diameters of 1–100 nm, are promising materials in the chemical and material sciences due to their essential properties. Recently, NPs have attracted great attention in the preparation of monolithic columns due to their amazing properties in relation to high specific surface area [13,14,15]. NPs incorporated into monoliths provide a large surface-to-volume ratio that can promote mass transfer as well as other promising properties such as high efficiency and stability. Among other NPs, fumed silica NPs (FSNPs), as a type of silica nanoparticle, have promising properties such as high surface area, good mechanical stability, and adsorption capability. These particles were used for the synthesis of the spin-tip columns for high-throughput lectin affinity chromatography of glycoproteins [16]. In recent reports, FSNPs were used for the preparation of monolithic columns for use in various analytical systems, such as HPLC [17,18,19], nano-LC [20], and CEC [21]. All these FSNPs are based on AEROSIL^®^ 200, which has a hydrophilic character in nature. AEROSIL^®^R972 is also a type of FSNP, which is obtained after being treated with dimethyldichlorosilane. These NPs have both highly dispersed nonporous morphology and highly hydrophobic character.

The advanced stationary phases in liquid phase separation techniques such as nano-LC are highly essential due to the fact that molecular omics separations are mostly performed using nano-columns with narrow inner diameters, operated at nL/min [7,9,13]. In this sense, the development of new monolithic nano-columns seems an important strategy for advanced omics applications. Monolithic columns are promising materials and can be easily prepared in small internal-diameter columns [10]. These types of nano-columns are quite an attractive option for the separation and analysis of limited-size samples [22]. Monolithic nanocolumns have great potential in order to improve sensitive separations and analyses [23].

In this study, we reported a new feature of HFSNPs for the preparation of new monolithic nanocolumns, which were prepared via an in-situ polymerization of BMA, ethylene dimethacrylate (EDMA) and HFSNPs in 50 µm i.d. fused silica capillaries. The parameters that affect the morphology of the column were systematically investigated during polymer synthesis. After preparation and characterization, the applicability of the final HFSNPs monolithic nano-column was demonstrated with a gradient separation of intact proteins in nano-LC.

## 2. Results and Discussion

### 2.1. The Monolithic Column Preparation and Characterization with 50 µm ID

In a very recent research report, BMA-based monolithic columns were prepared and applied for proteomics analysis by nano-LC [24]. Although this column demonstrated satisfactory performance, the polymerization content needs to be optimized due to new nanostructures (e.g., HFSNPs). Another essential issue for the preparation of new NPs incorporated monoliths is the selection of the porogenic solvent in which NPs are dissolved and homogenously dispersed. In this sense, the composition of the column was changed in order to prepare a new homogenous polymer solution with HFSNPs. In this preparation of the column, both BMA as the main monomer and EDMA as a crosslinker may impart both the support and the hydrophobic character to the surface.

The composition of the polymerization solutions used in the preparation of the monoliths is given in Table 1. The optimization of BMA and EDMA contents was achieved according to the published literature [24]. The selection of the solvent system was the first critical issue for both the solubility and the dispersion of HFSNPs into the polymerization solution, which is essential for the preparation of HFSNPs incorporated monoliths. In this sense, various solvent systems such as 2-propanol/dodecanol and cyclohexanol/dodecanol were used, and those solvent systems exhibited very low dispersibility. In light of these preliminary results, DMF and water were selected as the porogenic solvent systems, considering the HFSNPs solubility. It should be noted that the use of an ultrasound tip sonicator for this purpose allowed the homogeneous dispersion of the NPs.

Various contents of DMF in the porogenic solvent mixture were examined during the preparation of the HFSNP solution. It should also be noted that the water in the porogenic solvent was replaced with dodecanol. However, the monolithic structure did not show mechanical stability using the porogenic system with dodecanol, despite a homogenous polymerization mixture could be obtained. This could indicate that dodecanol is not a suitable solvent for HFSNPs. The best porogenic solvent ratio was found to be DMF/water with a 7:1 ratio (see Table 1). After optimization, the polymerization solution with HFSNPs, BMA, and EDMA could be well dissolved in the porogenic solvent containing DMF/water. Therefore, the homogenous polymerization solution could be obtained. Various contents of HFSNPs, such as 1.63, 4.94, 8.29, 13.4, 15.2, and 20.21 wt%, were used for the monolith preparation. Various amounts of HFSNPs on both the permeability and back pressure of the columns were also examined. When increasing the content of HFSNPs, the obtained monolith could not exhibit the desired permeability due to the higher content of HFSNPs that affected the polymerization process, and therefore a 13.4 wt% level with respect to the total polymerization mixture was found to be optimal.

Figure 1A,B shows the SEM images of the monolithic nano-column with HFSNPs (i.e., Column C7). The obtained results showed that HFSNPs were homogeneously distributed and incorporated into the structure.

SEM images revealed that both nanoglobules with 100 nm were obtained and the pore size was 3 μm. These results may prove the large surface area of a monolithic nano-column. Column 3 and Column 7 were also characterized by FT-IR spectroscopy (see Figure 2). 

According to the FT-IR spectrum, the enhanced absorption bands at 2958.22 cm^−1^ and at 1723.96 cm^−1^ were shown, while several others confirmed readily that the functionalization of the HFSNPs into the structure of the monolith. The pore size of the monoliths was measured using nitrogen physisorption, while the specific surface area of the columns area was calculated using the Brunauer–Emmett–Teller method. The surface area of Column 3 and Column 7 was calculated as 71.3 and 321.4 m^2^/g, respectively. The results showed that the specific surface area of the column with HFSNPs increased more than seven-fold when compared to the other one without HFSNPs.

Table 1 shows the hydrodynamic properties of the monolithic columns with or without HFSNPs. When ACN was applied as the mobile phase, Column 7 showed good permeability. The plot of the plate height versus the linear flow velocity (van Deemter plot) was examined by injecting a void marker (t_0_, thiourea) and ethylbenzene as a retained compound was used in the presence of a mobile phase consisting of ACN/H_2_O (i.e., 80/20%, *v*/*v*) (see Figure 3). The monolithic column synthesized with 13.4 wt% HFSNPs yielded a maximum plate height of 12.7 µm at optimum mobile-phase flow velocity, being able to generate more than ~85.000 plates/m. This result may indicate that HFSNPs-based monoliths allow rapid mass transfer.

The column back pressure (Δ*P*_column_) as a function of flow rate was examined in order to evaluate all column stabilities using ACN/H_2_O 80:20 (*v*/*v*) as the mobile phase. Figure 4 shows the optimized monolith (i.e., Column 7) stability, which exhibited a linear relationship (R^2^ = 0.9995) between the flow rate and the back pressure.

### 2.2. Chromatographic Evaluation and Application of HFSNPs-Based Monolithic Nanocolumn

#### 2.2.1. ABs Separation

Chromatographic characterization was performed using a homologous series of five alkylbenzenes (ABs) (e.g., toluene, ethylbenzene, propylbenzene, butylbenzene, and pentylbenzene). The ABs were chromatographed on the HFSNPs incorporated into monolithic nano-columns (i.e., C7). Various flow rates were applied for the separation of ABs using 85:15% *v*/*v* ACN:H_2_O as the mobile phase on Column 7, while column 3 (i.e., C3) gave no retention for ABs under the same chromatographic conditions. As shown in Figure 5A, satisfactory separation could be obtained at a flow rate of 400 nL/min.

In further optimization, various ACN contents were applied for ABs separation on Column 7. When decreasing the %ACN content from 85% to 65% in the mobile phase, the retention of ABs increased, leading to satisfactory peak shapes. Figure 5B shows promising separation of ABs homologous using the Column C7 by nanoflow LC.

The obtained results also obey Martin Equation [25], which is
Logk = (logα)n_c_ + logβ(1)
where n_c_ is the number of carbons in the alkyl chains of ABs, α is the methylene group selectivity increment, and β is the retention factor of the common residue (i.e., benzene) of the homologs. More details about the above equation, which is based on methylene group selectivity increment, were given in our previous published article [19].

Theoretical plate numbers up to 15,000 plates/m were readily obtained for ABs. When compared to a blank monolith without HFSNPs, the obtained results can be attributed to the fact that HFSNPs contain hydrophobic moieties and non-polar character. The stronger hydrophobic effect between ABs and the surface of HFSNPs makes them an ideal support for hydrophobic compounds. These results demonstrate that HFSNPs-based monolithic nano-column is a promising material for advanced chromatographic separations. Fresco-Cala et al. reported the preparation of a nanoparticle incorporated polymethacrylate monolith for small molecule separation [26]. In this study, the theoretical plate numbers up to 6000 plates/m could be obtained using ABs as test probes, while the present study exhibited better separation performance using HFSNPs-based monolithic nano-column.

#### 2.2.2. PAHs Separation

The prepared monolith was also used for PAH separation in terms of the hydrophobic character of HFSNPs. Six PAHs, including biphenyl, naphthalene, phenanthrene, anthracene, pyrene, and benzopyrene, were chromatographed on column 7 using ACN:H_2_O (80:20%, *v*/*v*) as the mobile phase. Figure 6A shows that the column exhibited promising baseline separation of PAHs. The results indicated that the hydrophobic character of HFSNPs led to strong hydrophobic interactions between the surface of the column and PAHs. Various ACN contents were also examined in the separation of PAHs using Column 7. Logarithmic retention factors vs. %ACN content of the mobile phase are given in Figure 6B. As shown here, the column retained PAHs by means of hydrophobic interactions. A linear correlation coefficient with a very high level (R^2^ > 0.995) could be achieved. These results confirm that the chromatographic retention mechanism of HFSNPs is based on reversed-phase behavior in nano-LC.

### 2.3. Protein Separation

Monolithic nano-columns are promising materials for molecular omics applications [27], where some challenges need to be tackled in proteomics analysis [28,29]. On the one hand, the relevant proteins are present at very low concentrations, which imposes great difficulties in analyzing these proteins. On the other hand, samples are usually obtained in extremely low volumes. In this sense, smaller i.d. columns may provide more sensitivity than large i.d. columns.

In this study, six standard proteins, including Cyt C (pI 10.6), RNase A (pI 9.6), Myoglobin (pI 6.8–7.3), Lys (pI 11.0), Ca isozyme II (pI 5.4) and α-Chymotryp A (pI 8.75) were initially used to evaluate the suitability of the monolithic nano-column for protein separation.

Linear gradient elution in ProFlow nano-LC was applied using a mobile phase containing 10% ACN/90% H_2_O at 0.1% *v*/*v* TFA as mobile phase A and 90% ACN/10% H_2_O at 0.1% *v*/*v* TFA as mobile phase B. The elution consisted of a linear gradient from 5–90% B in 8 min, followed by isocratic elution at 90% B for 5 min. This method was optimized according to our previous published article [20]. It can be seen in Figure 7 that six standard proteins were baseline separated using a 7 min linear gradient elution at increasing ACN concentration at a flow rate of 400 nL/min in ProFlow Nano-LC using an HFSNP-based monolithic nano-column.

The retention mechanism could be based on the strong hydrophobicity of the column originating from HFSNPs, which played an essential role in separating the proteins. Ma et al. prepared a Dipentaerythritol Penta-/Hexa-Acrylate Polyhidroxy polyhedral oligomeric silsesquioxane methacrylate (DPEPA-POSS) hybrid monolithic column with 100 µm i.d. applied for the separation of various proteins [29]. A better or comparable separation performance for the protein separation could be achieved.

### 2.4. Stability and Repeatability

Three HFSNP monoliths with 50 µm ID, which were prepared under the same chemical conditions, were prepared and evaluated for run-to-run, day-to-day, and column-to-column reproducibility. Both the retention times and the peak areas were assessed using RSD values. Table 2 shows the reproducibility values of the columns based on run-to-run, day-to-day, and column-to-column. The values were found as 2.30%, 2.48%, and 2.99% (*n* = 3), respectively. The HFSNP monolithic column showed satisfactory results with reproducible retention times using the retention factor of toluene as the test compound (thiourea as void marker). The resulting RSD values were indicative of the fabrication of the HFSNP monolithic column.

## 3. Materials and Methods

### 3.1. Instrumentation

Butyl methacrylate (BMA) (99%), ethylene dimethacrylate (99%) (EDMA) were purchased from Merck. AEROSIL^®^R972 nanoparticles (Lot no: 150012923) from Evonik were obtained as a gift by the company (Marmara Ecza ve Kimyevi Maddeler, Istanbul, Turkey). Polyaromatic hydrocarbons (PAHs) including biphenyl, naphthalene, anthracene, phenanthrene, pyrene, and benzopyrene were purchased from Merck A.G (Darmstadt, Germany). Fused silica capillaries with 50 µm ID and 363 µm OD (Lot: BUHT02A) were obtained from BGB Analytik. A reducing union 1/16′’ to 360 µm was purchased from VICI Valco in order to connect capillary. All standard proteins including ribonuclease A (RNase A), cytochrome C (Cyt C), Ca isozyme II, lysozyme (Lys), myoglobin (Mb), α-chymotrypsinogen A (α-Chymotryp A) were purchased from Sigma Aldrich (St. Louis, MO, USA).

LC-UV experiments were carried out using ProFlow-nano-LC from Thermo Fischer Scientific Dionex Ultimate 3000 Series (Waltham, MA, United States). ProFlow Nano-LC includes a solvent rack (SRD-3400), a NCP-3200RS pump, and a variable wavelength UV-Vis detector-3400RS with a 3 nL flow cell as well as an autosampler with WPS-3000TPL RS. Pure water was obtained using an ultrapure water system (Direct-Q^R^-3) from Millipore Corp. (Billerica, MA, USA). This system incorporates an LC pack solution part (LC-Pak^R^ Polisher Catalogue No LCPAK0001) (Billerica, MA, USA). In order to the homogeneous dispersion of AEROSIL^®^R972 (HFSNPs) into the porogenic solvent, a tip ultrasound sonicator (BANDELIN Electronic GmbH Co. KG Hsinrichstraβe 3–4 12,207 Berlin, Germany) was used.

### 3.2. Monolithic Nano-Column Preparation

The fused silica capillary with 50 µm i.d. was silanized according to the published article [20]. The polymeric solution was prepared using HFSNPs (13.4 wt%), BMA (10.2 wt%) as the monomers, EDMA (20.4 wt%) as a crosslinker, DMF (48.8 wt%) and water (7.04 wt%) as porogens, and AIBN (0.8 wt%) as initiator. Table 1 shows the content of polymerization mixture used for monolith preparation. The obtained mixture was stirred for 5 min and degassed in an ultrasonic bath for 5 min at 23 °C. The final mixture was injected into the silanized capillary with 12 cm length using a 0.5 mL syringe in order to prepare monolithic column. After capillary filling with polymerization mixture, the inlet- and outlets of the capillary were sealed with septa. The filled capillary was placed in a water bath for 18 h at 65 °C. In order to examine the effect of HFSNPs, which is named “blank monolith”. The above-mentioned procedure was also applied to the preparation of a monolithic column without HFSNPs under the same conditions. After polymerization, the monolithic columns were washed with ACN:H_2_O (80:20% *v*/*v*) for 2 h at a flow rate of 800 nL/min. These columns were further applied for the test solutes in nano-LC. In order to use for the characterization studies, including FT-IR and the measurement of specific surface area, the same polymerization mixtures were also prepared at a larger scale in 2 mL Eppendorf tubes.

### 3.3. Chromatographic Conditions

ABs and PAHs were dissolved in ACN and then diluted with deionized water. Both standard ABs and PAHs solutions were in the range of 0.01–200 µg/kg while the protein standard solutions were prepared in the range of 0.1–500 µg/kg. These stock solutions were prepared and stored in the refrigerator with shelf-life of 20 days.

Standard protein separation was obtained by a gradient elution with mobile phase A: 10%ACN/90%H_2_O at 0.1% *v*/*v* TFA; mobile phase B, 90% ACN/10% H_2_O at 0.1% *v*/*v* TFA; linear gradient elution, 5–90% B in 8 min, followed by isocratic elution at 90% B for 5 min. The flow rate was in the range of 200–800 nL/min in the ProFlow nano-LC system.

## 4. Conclusions

In this study, HFSNPs-based monolithic nano-columns were prepared and applied for the separation of small molecules and protein separations. The column was optimized according to its HFSNP content. The various chromatographic conditions were evaluated using the optimized column. The proposed monolith with a 50 µm ID offered promising separations for both small molecules (e.g., alkylbenzenes and polyaromatic hydrocarbons) and macromolecules (e.g., six intact proteins). The separation mechanism was based on the strong hydrophobicity of HFSNPs incorporated into the monolithic column. The developed monolithic nano-column demonstrated good separation performance for small molecules and intact proteins. The developed HFSNPs-based monolithic nano-columns are promising for nano-LC, especially for proteomics applications.

## Figures and Tables

**Figure 1 molecules-27-02306-f001:**
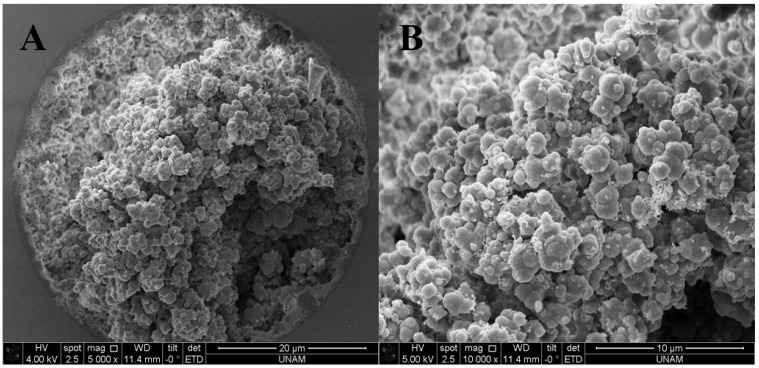
SEM images of the monolithic nano-column with magnification (**A**) ×5000 (**B**) ×10,000 (**A**,**B**).

**Figure 2 molecules-27-02306-f002:**
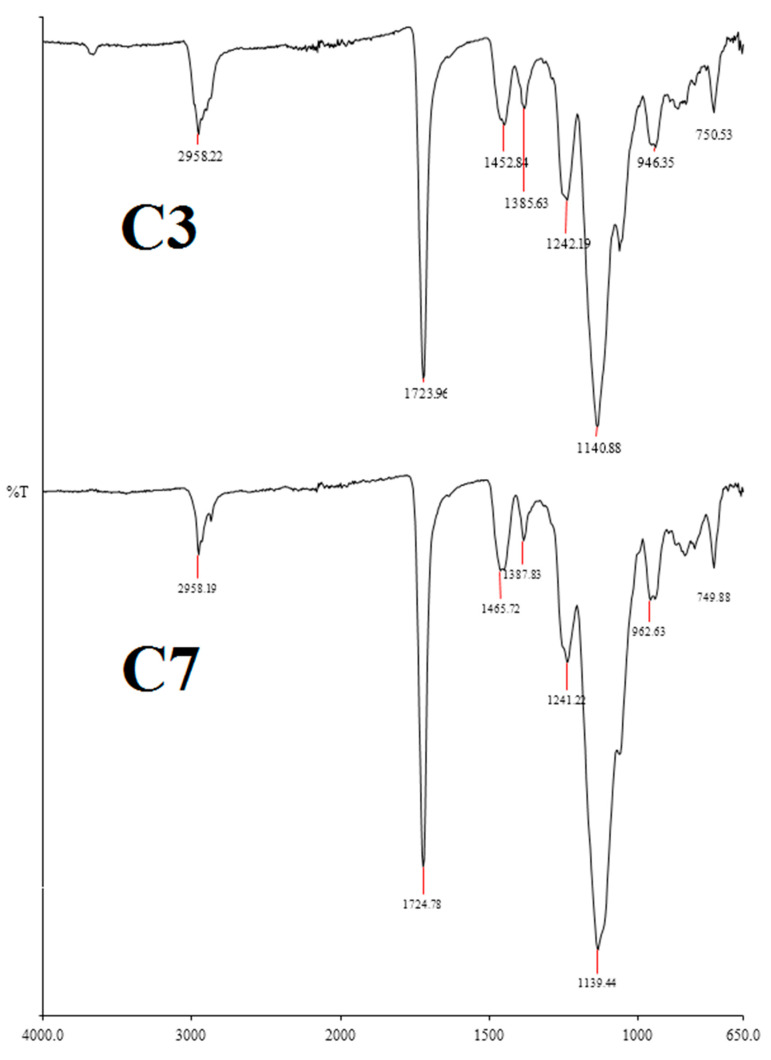
FT-IR spectra of the columns: Column 3 and Column 7.

**Figure 3 molecules-27-02306-f003:**
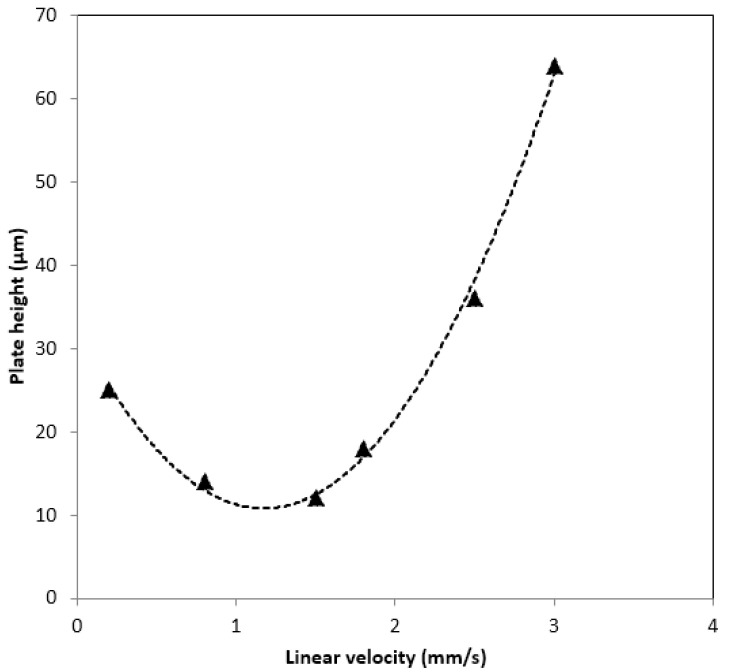
Van Deemter plot for ethylbenzene as a retained compound using the monolithic column.

**Figure 4 molecules-27-02306-f004:**
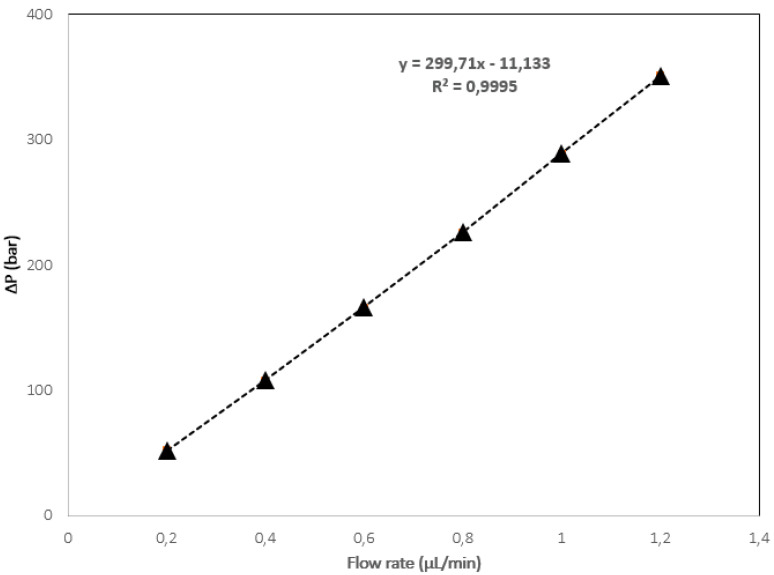
Plot of the monolithic column with 50 µm i.d. back pressure versus flow rate with a mobile phase composed of (ACN/H2O (80:20, *v*/*v*%).

**Figure 5 molecules-27-02306-f005:**
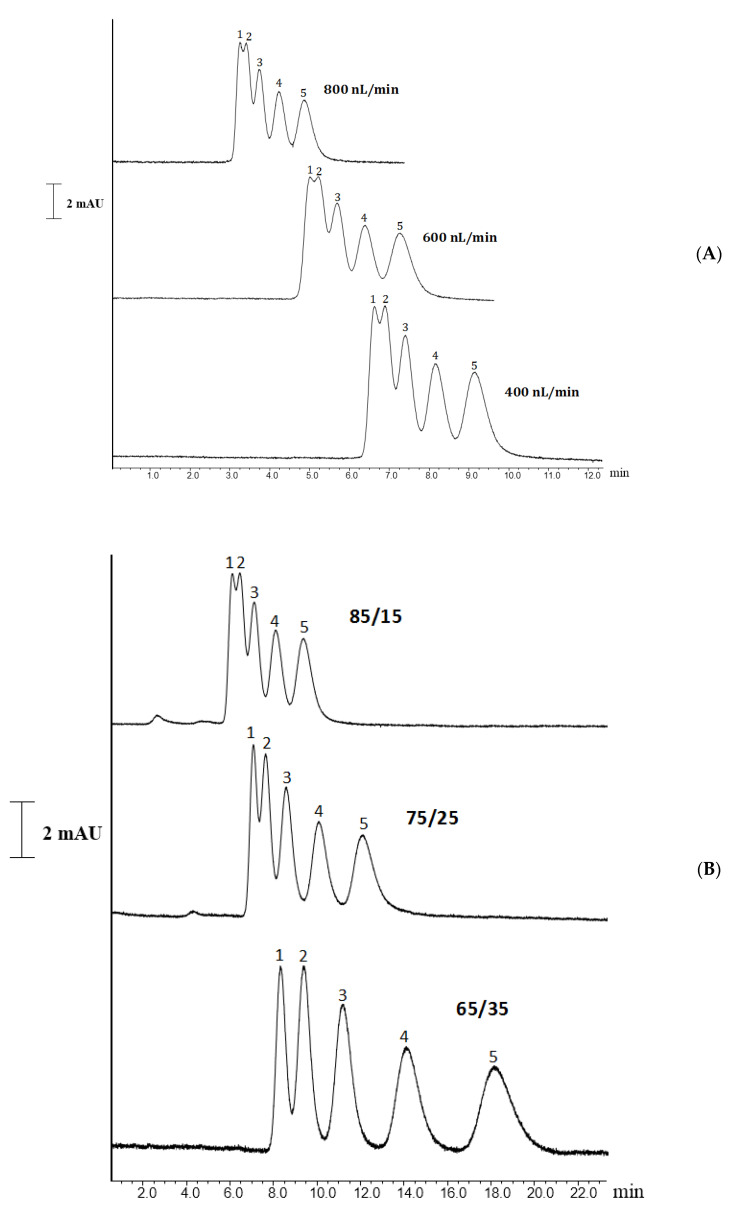
(**A**) Separation chromatograms of alkylbenzenes on Column 7 at various flow rates with 800 nL/min, 600 nL/min and 400 nL. Chromatographic conditions: mobile phase, 80%ACN:20%H_2_O (*v*/*v*); column dimensions, 13 cm × 50 µm i.d.; analyte concentration, 0.05 µg/mL; detection wavelength, 220 nm. Order of peaks: (1) toluene (2) ethylbenzene, (3) propylbenzene, (4) butylbenzene, (5) pentylbenzene. (**B**) Separation chromatograms of alkylbenzenes on Column 7 at various ACN content: 85/15% (*v*/*v*), 75/25% (*v*/*v*), and 65/35% (*v*/*v*). Column dimensions, 13 cm × 50 µm i.d.; analyte concentration, 0.05 µg/mL; flow rate, 400 nL/min; detection wavelength, 220 nm. Order of peaks: (1) toluene (2) ethylbenzene, (3) propylbenzene, (4) butylbenzene, (5) pentylbenzene.

**Figure 6 molecules-27-02306-f006:**
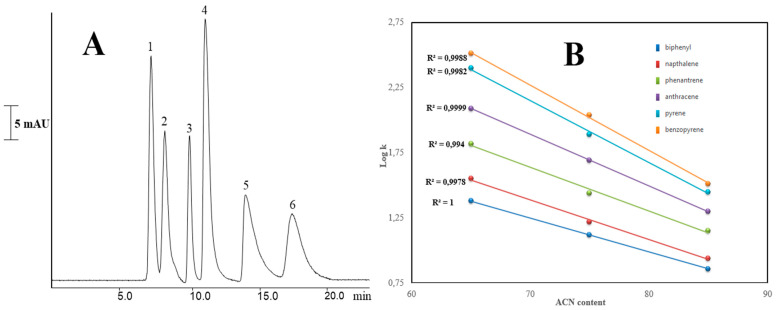
(**A**) Separation chromatogram of PAHs on Column 7. Chromatographic conditions: mobile phase. 80% ACN:20% H_2_O (*v*/*v*); column dimensions, 13 cm × 50 µm i.d.; analyte concentration, 0.03 µg/mL; detection wavelength, 220 nm. Order of peaks; (1) biphenyl (2) naphthalene, (3) phenanthrene, (4) anthracene, (5) pyrene, (6) benzopyrene. (**B**) The effect of ACN content on retention of PAHs using Column 7. The other conditions are the same as in (**A**).

**Figure 7 molecules-27-02306-f007:**
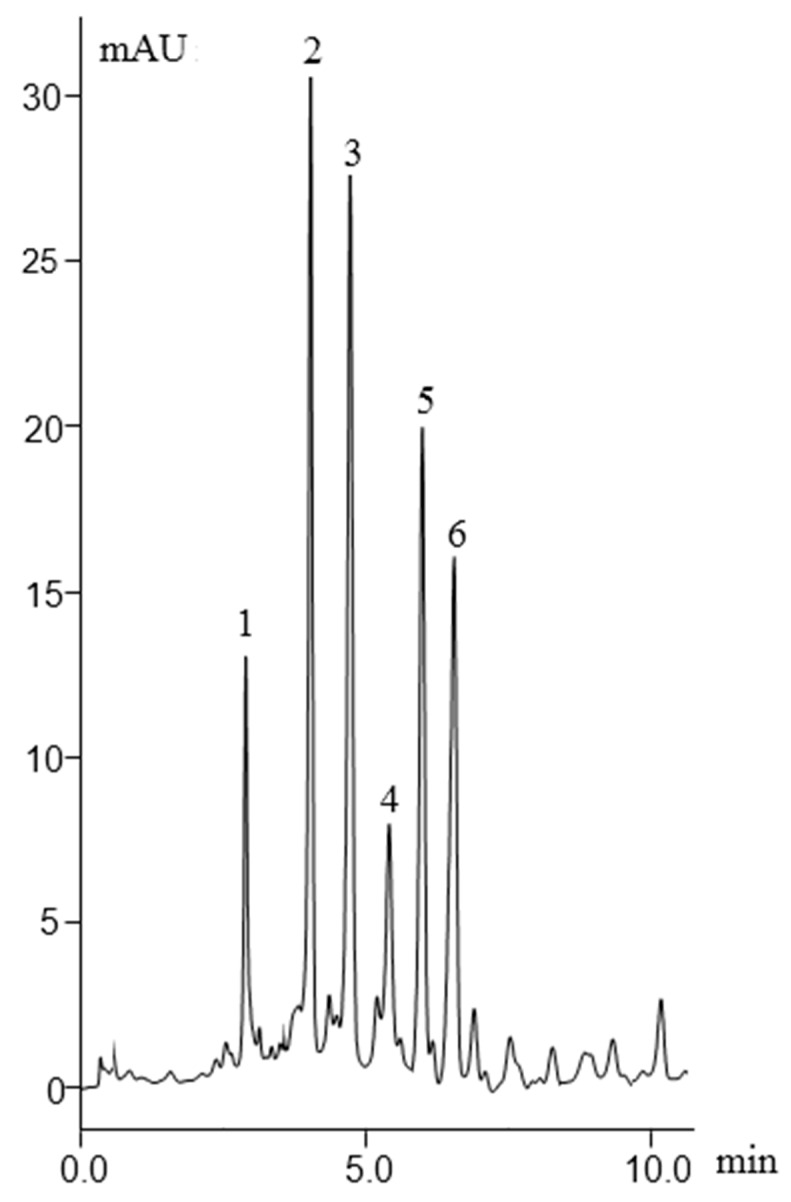
Separation chromatogram of intact proteins using C7 column. Mobile phase A, 10%ACN/90%H_2_O at 0.1% *v*/*v* TFA; mobile phase B, 90% ACN/10% H_2_O at 0.1% *v*/*v* TFA; linear gradient elution from 5–90% B in 7 min, followed by isocratic elution at 90% B for 5 min. Detection wavelength, 214 nm. Peaks order, (1) RNase A, (2) Cyt C, (3) Ca isozyme II, (4) Lys, (5) Mb (6) α-chymotrypsinogen A.

**Table 1 molecules-27-02306-t001:** Composition of the polymerization solutions used in the preparation of the monolithic columns with or without HFSNPs.

Column	Monomers (µL)		Porogen (µL)	R^2 b^	Specific Surface Area (m^2^/g)	Permeability (m^2^) (×10^−14^)
	BMA:EDMA	HFSNPs ^a^	DMF:H_2_O ^c^			
C1	104:66	-	352:48	No back pressure	-	-
C2	104:76	-	352:48	No back pressure	-	-
C3	104:86	-	352:48	0.9990	71.3	9.14
C4	104:86	1.63	352:48	0.9995	-	-
C5	104:86	4.94	352:48	0.9997	-	-
C6	104:86	8.29	352:48	0.9995	-	-
C7	104:86	13.4	352:48	0.9999	321.4	1.48
C8	104:86	15.2	352:48	-	-	-
C9	104:86	20.2	352:48	-		-

- no calculation. ^a^ Percentage of HFSNPs in the total polymer mixture (wt%). ^b^ Linear relationship between flow rate and the resulting backpressure of relevant monolith. ^c^ The ratios between DMF and H_2_O as porogens.

**Table 2 molecules-27-02306-t002:** The stability of retention behaviour of the prepared HFSNP monolithic column with 50 µm ID at 80/20% ACN/H_2_O flow rate; 400 nL/min, 12 cm × 20 µm i.d., concentration; 0.2 µg/mL, detection wavelength; 220 nm.

Type and Number (*n*) of Experimental Repetitions	Average Retention Time (min)	RSD% of Retention Time
Run-to-run (*n* = 3)	2.15	2.30
Day-to-day (*n* = 3)	2.21	2.48
Column-to-column (*n* = 3)	2.39	2.99

## Data Availability

The data presented in the current study are available in the article.

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
