# Peer review of "Hydrophobic AEROSIL®R972 Fumed Silica Nanoparticles Incorporated Monolithic Nano-Columns for Small Molecule and Protein Separation by Nano-Liquid Chromatography"

_molecules, 2022, doi:10.3390/molecules27072306_

Round 1

Reviewer 1 Report

The paper systematically described the development of monolithic nano-column and demonstrated its application for small molecule and protein separation. 

Although the authors clearly characterized the columns they described, it would be of interest to the readers how the described column compares to other existing columns, commercially available or published paper on monoliths with nano particles. Is this developed column better than those columns? If yes, how? If no, what could be improved? 

The authors also mentioned several times about mechanical stability however, there is no study showing the stability of the column. Did the authors used the column multiples times? Could the authors provide data showing the stability of separation efficiency and peak shape over several runs? 

Citation for article 18 and 19 are the same, please edit your citations.

Author Response

Comment -1

The paper systematically described the development of monolithic nano-column and demonstrated its application for small molecule and protein separation. 

Response-1

The authors are thankful for editor’s positive assessment of the manuscript. The original manuscript was revised according to reviewer comments.

Comment-2

Although the authors clearly characterized the columns they described, it would be of interest to the readers how the described column compares to other existing columns, commercially available or published paper on monoliths with nano particles. Is this developed column better than those columns? If yes, how? If no, what could be improved?

Response-2

The developed column was compared with the other existing column and this was added to section 2.2.1 in the revised manuscript.

Comment-3

The authors also mentioned several times about mechanical stability however, there is no study showing the stability of the column. Did the authors used the column multiples times? Could the authors provide data showing the stability of separation efficiency and peak shape over several runs? 

Response-3

Fumed silica nanoparticles show good mechanical stabilities. The relevant properties were also shown in the monolithic column. Fig. 4 demonstrated the mechanical stability of the monolith.  

Comment-4

Citation for article 18 and 19 are the same, please edit your citations.

Response-4

It was corrected.

Reviewer 2 Report

1- for how many times a column can be used under the same conditions and for the same analytes (please be specific on the plate numbers, if possible)

2- Can you prepare a comparison table to shoe the advantages of your column to the similar studies?

3- It is recommended to add a SEM of the column after several separation

Author Response

Comment -1

for how many times a column can be used under the same conditions and for the same analytes (please be specific on the plate numbers, if possible) 

Response-1

The related part was given in section 2.4

Comment-2

Can you prepare a comparison table to shoe the advantages of your column to the similar studies?

Response-2

This monolithic column is with 50 µm ID. According to our acknpwledge, it is difficult to prepare packed based column with 50 µm ID and  there is no the column like that and that is why the article contribute  literature greatly.

Comment-3

It is recommended to add a SEM of the column after several separation

Response-3

This is a good comment but currently SEM instrument is not available.

Reviewer 3 Report

Well-written manuscript, easy to read, and well-worth publishing. I support its publication. I have one comment for the Authors to consider: The Authors recommended the column for proteomics application. I agree with this conclusion, but in my opinion the mauscript would be much better if the Authors show an axample of peptide separation (possibly use it in a bootom up proteomics experiment). 

Minor comment:

line 301 "tried" should be deleted

Author Response

Comment -1

Well-written manuscript, easy to read, and well-worth publishing. I support its publication. I have one comment for the Authors to consider: The Authors recommended the column for proteomics application. I agree with this conclusion, but in my opinion the mauscript would be much better if the Authors show an axample of peptide separation (possibly use it in a bootom up proteomics experiment). 

Response-1

The authors are thankful for editor’s positive assessment of the manuscript. Regarding proteom application, it would be another topic for deep investigation.

Reviewer 4 Report

It looks like there is a problem with some special characters ("@" instead of alpha or beta, I suppose) - please check.

Author Response

Comment -1

It looks like there is a problem with some special characters ("@" instead of alpha or beta, I suppose) - please check.

Response-1

Thank you. It was revised as suggested.

Reviewer 5 Report

Comments

The authors fabricate a nano-column using narrow monolithic fused silica capillary column modified with hydrophobic fumed silica nanoparticles (HFSNPs). They investigated the prepared column for separating some alkylbenzenes, polyaromatic hydrocarbons and proteins separation. The work carried out by the authors is duly acknowledged. However, please consider the following essential recommendations and suggestions in order to improve the quality of your manuscript and to be suitable for publication at the Journal of Molecules.

  • In table 1, state the unit of the permeability and why it is calculated only for C3 and C7?
  • what is “d” letter kept at DMF:H2Od in table 1?
  • the used van Deemter equation should be stated in the manuscript and the parameters that are used in calculating the plate high should be mentioned.
  • Page # 8, What is the impact of various ACN percentage in the mobile phase on the response and peak broadening?
  • The separation optimization parameters should be calculated and stated; Separation Efficiency, Resolution, distribution constant, Retention Factor and Selectivity.

Author Response

Comment -1

In table 1, state the unit of the permeability and why it is calculated only for C3 and C7?

Response-1

It was revised as suggested. C3 is plain monolith while the other one  (C7) was the optimized monolith according to HFSNP content.

Comment-2

what is “d” letter kept at DMF:H2Od in table 1?

Response-2

It was revised as suggested

Comment-3

for how many times a column can be used under the same conditions and for the same analytes (please be specific on the plate numbers, if possible)

Response-3

Kindly see section 2.4.

Comment-4

the used van Deemter equation should be stated in the manuscript and the parameters that are used in calculating the plate high should be mentioned.

Response-4

It is a general equation and can be found many literatures (e.g. see ref. 3) .

Comment-5

Page # 8, What is the impact of various ACN percentage in the mobile phase on the response and peak broadening?

Response-5

It is just based on hydrophobic interactions.

Comment-6

The separation optimization parameters should be calculated and stated; Separation Efficiency, Resolution, distribution constant, Retention Factor and Selectivity.

Response-6

It can be seen from figures 5A-B. Generally, those parameters or chromatograms may be given in an article.

Round 2

Reviewer 5 Report

The authors addressed the comments. Check the number of Tables in the revised manuscript.